# Impact of Rhamnolipids (RLs), Natural Defense Elicitors, on Shoot and Root Proteomes of *Brassica napus* by a Tandem Mass Tags (TMTs) Labeling Approach

**DOI:** 10.3390/ijms24032390

**Published:** 2023-01-25

**Authors:** Elise Pierre, Paulo Marcelo, Antoine Croutte, Morgane Dauvé, Sophie Bouton, Sonia Rippa, Karine Pageau

**Affiliations:** 1Unité Transfrontalière BioEcoAgro, BIOlogie des Plantes et Innovation (BIOPI), UMRt 1158, Université de Picardie Jules Verne, 80039 Amiens, France; 2Plateforme d’Ingénierie Cellulaire & Analyses des Protéines ICAP, FR CNRS 3085 ICP, Université de Picardie Jules Verne, 80039 Amiens, France; 3Unité de Génie Enzymatique et Cellulaire, UMR CNRS 7025, Alliance Sorbonne Universités, Université de Technologie de Compiègne, 60203 Compiègne, France

**Keywords:** biological control, quantitative proteomics, rapeseed, rhamnolipids, TMT labeling

## Abstract

The rapeseed crop is susceptible to many pathogens such as parasitic plants or fungi attacking aerial or root parts. Conventional plant protection products, used intensively in agriculture, have a negative impact on the environment as well as on human health. There is therefore a growing demand for the development of more planet-friendly alternative protection methods such as biocontrol compounds. Natural rhamnolipids (RLs) can be used as elicitors of plant defense mechanisms. These glycolipids, from bacteria secretome, are biodegradable, non-toxic and are known for their stimulating and protective effects, in particular on rapeseed against filamentous fungi. Characterizing the organ responsiveness to defense-stimulating compounds such as RLs is missing. This analysis is crucial in the frame of optimizing the effectiveness of RLs against various diseases. A Tandem Mass Tags (TMT) labeling of the proteins extracted from the shoots and roots of rapeseed has been performed and showed a differential pattern of protein abundance between them. Quantitative proteomic analysis highlighted the differential accumulation of parietal and cytoplasmic defense or stress proteins in response to RL treatments with a clear effect of the type of application (foliar spraying or root absorption). These results must be considered for further use of RLs to fight specific rapeseed pathogens.

## 1. Introduction

Disease and pests represent major problems for sustainable agriculture in the world and cause severe yield damage, resulting in huge economic losses [1]. In this context, plants have developed defense mechanisms to counteract devastating pathogens. Plants possess morphological, physiological and biochemical defense mechanisms that constitute the innate immune response. In addition to their constitutive defenses such as preformed cuticular waxes and tough cell wall [2], plants can perceive pathogens and to set up so-called induced defenses. This induced resistance is activated through intracellular signaling that initiates early responses such as oxidative burst, which results in the production of a large amount of reactive oxygen species (ROS) [3] and late responses such as transcriptional changes and callose deposits [4,5]. Other slower reactions include the production of antimicrobial compounds, alteration of the plant cell wall and de novo synthesis of defense proteins [6].

Rapeseed (*Brassica napus* L.) is a crop of great economic interest being the most produced oilseeds in France and in the European Union and the second oilseeds just behind soybean in the world [7]. Rapeseed is grown for its seeds rich in oils and vegetable proteins used for human nutrition, animal feed and to a lesser extent, for fuels and technical oils [8]. However, rapeseed is exposed to many pathogens impacting seed yields [9]. To manage and fight rapeseed diseases, chemipesticides and fertilizers are routinely used by farmers, but they cause huge amount of environmental pollution and deleterious effects on health [10]. Development of alternative strategies such as biocontrol to reduce the use of synthetic pesticides for crop protection is becoming a necessity. Exploring the potential of natural elicitors to limit pathogen development and/or strengthen plant defense mechanisms has been studied for many years [11]. These elicitors, also known as Plant Defense Stimulators (PDS), increase plant tolerance in the event of an attack by a bioaggressor [12]. Compounds that have shown good potential as PDSs are rhamnolipids (RLs), glycolipids naturally produced and secreted by *Pseudomonas* species, mainly by the opportunistic pathogen *Pseudomonas aeruginosa*, and by some *Burkholderia* species [13,14]. These amphiphilic compounds are composed of one or two hydrophobic alkyl chains linked through a glycosidic bond to one (mono-RLs) or two (di-RLs) [15] and they have many biological properties and represent a great interest as biosourced and biodegradable surfactants [16]. Thanks to the properties of natural RLs, many industrial applications are being developed such as bioremediation of polluted soils, pharmaceutical formulations, cosmetics and more recently in agriculture for the control of plant pathogens [17,18]. RLs have direct antimicrobial properties against phytopathogens [13]. As elicitors they also stimulate plant innate immunity [19] in grapevine, *Arabidopsis thaliana* or *B. napus*. To date, a large range of RLs concentrations have been used to induce immunity on these various plant species without deleterious effect on their growth and their physiology [20,21,22]. Moreover, foliar applications of RLs obtained from *P. aeruginosa* also induce a local resistance against *Botrytis cinerea* and the hemibiotrophic fungus *Leptosphaeria maculans* in the early stages of infection in *B. napus* [22,23].

Given the amphiphilic nature of RLs, it has been suggested that these molecules could directly interact with the membrane lipids [21,24,25]. The direct antifungal properties of the RLs could be due to lysis of mycelial cells via the destabilization of membranes [13]. It should be noted that the lipid composition of the membranes could influence the effect of the RLs on the mycelial cell destabilization [26,27]. It has been shown that thanks to their amphiphilic nature, RLs are able to fit into plant lipid-based membrane models and are located near the lipid phosphate group of the phospholipid bilayers, nearby phospholipid glycerol backbones [24]. This insertion would result in structural changes to the membranes, without affecting lipid dynamics and membrane fluidity. Moreover, it is not clear whether the RL-triggered protection is driven by activation of plant defense responses and/or antimicrobial properties. This protection is associated with a higher induction of defense-related enzymes and the accumulation of antimicrobial metabolites [28,29,30,31]. Bacterial RLs are therefore an interesting alternative to the use of conventional plant protection products due to their antimicrobial and defense eliciting properties [32]. Their low toxicity and biodegradability are also significant assets for their use in agriculture [33]. However, information about the effect of the mode of application of RLs is missing to better evaluate their potential to protect plants from pathogens.

To our knowledge, there is no data reporting the characterization of the global proteomic response of *B. napus* after a RL treatment by foliar spraying or root application. In this study, a comparative quantitative proteomic analysis has been performed to investigate the difference between the protein profiles of shoots and roots of *B. napus* under the two types of RL applications, during 7 h and 24 h.

## 2. Results

### 2.1. RLs Significantly Modify Protein Abundance in Rapeseed Shoots and Roots upon Elicitation

We performed a comparative quantitative proteomic analysis to investigate the plant responses to 2 types of RL applications during 2 time points (T 7 h and T 24 h) in shoots and roots of rapeseed seedlings. RL treatments consisted of foliar spraying or root absorption of 0.1 g L^−1^ RL solutions, this concentration being known to trigger defense mechanisms in rapeseed [23]. Our objectives were to investigate changes in protein abundance caused by RLs on treated tissues (local responses) and/or on untreated tissues (systemic responses).

Our results showed differences in protein abundance upon RL treatment in shoots and roots of rapeseed. Altogether, 906 were Differentially Accumulated Proteins (DAPs), identified according to a fold change (FC) >1.7 or <0.6 with a *p*-value < 0.05 in shoots and roots regardless of the type of treatment and time points (Figure 1). Among them, 77 proteins were under-accumulated in shoots whereas 387 were under-accumulated in roots after RL treatment by foliar spaying or root application at 7 h and 24 h. The same trend was found for over-accumulated proteins with 172 proteins in shoots and 270 in roots, albeit with a smaller margin (Figure 1A, 1B). These results highlighted the impact of RL treatment on the different parts of *B. napus* with a stronger effect on roots.

Specifically, in shoots, we found only 4 DAPs over-accumulated after 7 h of treatment by foliar spraying (Figure 1A) whereas after 24 h of treatment by foliar spraying, 42 DAPs were identified (12 DAPs under-accumulated and 30 DAPs over-accumulated). After RL application on roots, we found 59 (20 DAPs under-accumulated and 39 DAPs over-accumulated) and 144 DAPs (45 DAPs under-accumulated and 99 over-accumulated) at, respectively, T 7 h and T 24 h in shoots. Thus, RL application on roots had a larger impact on protein abundance in shoots compared to foliar spraying. Interestingly, modifications of protein abundance were already greater after 7 h of RL treatment by root absorption (59 DAPs) compared to foliar spraying at T 24 h (42 DAPs), further highlighting the minor local response induced by foliar spray of RLs in shoots.

In roots, 34 DAPs (14 under-accumulated and 20 over-accumulated) and 63 DAPs (48 under-accumulated and 15 over-accumulated) were found after RL application by foliar spray at T 7 h and T 24 h, respectively (Figure 1B). Foliar spraying of RLs had a larger impact on root protein abundance than on shoot protein abundance (Figure 1A,B). Additionally, we identified 268 (165 under-accumulated and 103 over-accumulated) and 292 DAPs (160 under-accumulated and 132 over-accumulated) at, respectively, T 7 h and T 24 h after RL application by root absorption, thus underlining an extensive local response in roots. Interestingly, protein abundance at T 7 h and T 24 h after RL treatment by root absorption were quite similar, with an increase of only 24 DAPs at T 24 h compared to T 7 h. This result is clearly different in comparison with the impact of RL application by root absorption on shoot protein abundance where we found a strong increase of 85 DAPs at T 24 h compared to T 7 h (Figure 1A). Globally, we found more under-accumulated proteins than over-accumulated proteins in roots, which is in contrast with what was found in shoots where more proteins were over-accumulated (Figure 1).

### 2.2. Mode of RL Application Differently Influences Protein Accumulation at the Local and Systemic Level in Rapeseed

We then assessed the proteins which were differentially accumulated across multiple treatment conditions to determine the specificity of the plant response to RLs depending on the type of application (Figure 2). 

At T 7 h, there were no shared DAPs between shoots and roots treated with RLs by foliar spraying (Figure 2A). At T 24 h, only 1 protein was differentially accumulated in both shoots and roots upon foliar spray treatment (Figure 2A). This protein which was under-accumulated in both shoots and roots (Appendix A) corresponds to the heat shock protein (HSP) 90-2-like (A0A078GD98).

After RL application by root absorption, 14 proteins were differentially accumulated in both shoots and roots at T 7 h (Figure 2B). Thirteen out of these 14 DAPs were over-accumulated in both shoots and roots. Logically, most DAPs had a higher abundance ratio in roots as we previously showed an extensive local response in roots. Most abundant DAPS included the farnesoic acid carboxyl-O-methyltransferase-like (A0A078JEJ1) (FC of 17.38 in roots against 2.84 in shoots) and the phenylalanine ammonia-lyase (PAL; A0A078JLX5) (FC of 8.13 in roots against 1.93 in shoots) (Appendix A). The only under-accumulated protein in both organs was a high mobility group nucleosome-binding protein (A0A078JRJ2) (FC of 0.50 in roots against 0.59 in shoots) (Appendix A). At T 24 h, we identified 17 shared DAPs between shoots and roots treated with RLs by root absorption (Figure 2B). Interestingly, we found that 6 over-accumulated proteins in shoots were under-accumulated in roots and that 2 under-accumulated proteins in shoots were over-accumulated in roots (Appendix A). The rest of the shared DAPs were over-accumulated in both organs.

### 2.3. Plant Defense/Stress, Transport and Secondary Metabolism Proteins Are DAPs Most Widely Represented in Shoots and Roots of Rapeseed upon RL Treatment at T 7 h

To better evaluate differences between local and systemic responses upon different types of RL treatment in rapeseed shoots and roots, the DAPs were distributed into functional categories according to their putative functions. Across all conditions, we could sort the DAPs into 19 functional categories (Figure 3 and Figure 4; Appendix A).

In shoots, only 4 over-accumulated proteins were identified in response to RL application by foliar spray at T 7 h (Figure 3B). Two of them were involved in photosynthesis and photorespiration, and the other two were involved in transport and transcription/translation processes (Appendix A). No proteins were found to be under-accumulated in shoots treated by foliar spray during 7 h. Contrastingly, 39 and 20 proteins were, respectively, over- and under-accumulated in shoots treated with RLs by root absorption. Most over-accumulated proteins were involved in secondary metabolism (10 proteins; 26%) and plant defense (4 proteins; 10%). Among the DAPs related to secondary metabolism, we identified several PALs, glycosyltransferases (GTFs) which accumulated around 2-fold versus control and an isoflavone reductase homolog P3 (A0A078I5Q4) which accumulated by 4.2-fold versus control (Appendix A). Transport was the most represented category among under-accumulated proteins (5 proteins; 25%) with other categories evenly represented with 1 or 2 proteins each.

In roots, 20 over-accumulated and 14 under-accumulated proteins were identified when RLs were applied by foliar spraying on shoots at T 7 h (Figure 3B). Plant defense/stress was the most represented functional category for both over-accumulated (5 proteins; 25%) and under-accumulated proteins (3 proteins; 21%). Putative functions of plant defense DAPs included response to both abiotic stress and biotic stress. Additionally, we described 5 over-accumulated proteins related to transport, though the most abundant protein was a feruloyl CoA ortho-hydroxylase 1-like (A0A078GB10) involved in secondary metabolism and over-accumulated by 6-fold (Appendix A). Other under-accumulated proteins were linked to photosynthetic processes, DNA/RNA binding, transcriptional and transitional processes, and protein modification or degradation (Appendix A). 

As previously mentioned, DAPs were more abundant in roots when RLs were applied locally on those organs as we found 103 over-accumulated and 165 under-accumulated proteins (Figure 3B). As with roots from seedlings treated by spraying of RLs on leaves, plant defense/stress was the most represented category with 18 over-accumulated proteins (17%) and 26 under-accumulated proteins (16%). Among the DAPs involved in abiotic stress responses, most of them had a putative function related to response to cold/heat or drought stresses such as HSPs and late embryogenesis abundant proteins (LEA) (Appendix A). DAPs linked to biotic responses included pathogenesis-related proteins (PR1-like protein [A0A078FFW4]; PR4-like protein [A0A078HV79]), enzymes with chitinase activity and ankyrin repeat-containing proteins (Appendix A). Other major functional classes in over-accumulated proteins included secondary metabolism with 17 proteins (17%) and DNA/RNA binding/transcription/translation/protein folding with 11 proteins (11%) (Figure 3B). Among DAPs participating in secondary metabolism, the majority were related to flavonoids and glucosinolates metabolism. Enzymes which were exclusively over-accumulated after root absorption were PALs and GTFs (Appendix A). Moreover, five over-accumulated proteins associated with phytohormones and signaling pathways were identified, such as a 1-aminocyclopropane-1-carboxylate oxidase-like (ACC oxidase; A0A078FIY9), a 12-oxophytodienoate reductase 1 (OPR; A0A078I7U4) and a farnesoic acid carboxyl-O-methyltransferase-like (FAMT; A0A078JEJ1) (Appendix A). The OPR and the FAMT were also over-accumulated in shoots after RL application by root absorption (Appendix A).

Beside plant defense proteins, we identified under-accumulated transport proteins (26 proteins; 16%), DNA/RNA binding and transcription/translation proteins (24 proteins; 15%), ROS scavenging enzymes/redox homeostasis proteins and signal transduction proteins (both at 9 proteins; 5%). Eight proteins related to lipid metabolism were DAPs in roots at T 7 h after root absorption treatment, which included proteins from the GDSL (motif consensus amino acid sequence of Gly, Asp, Ser, and Leu) esterase/lipase family, phospholipid metabolism and fatty acid metabolic processes (Appendix A).

### 2.4. RLs Modify More Diverse Functional Protein Categories in Rapeseed upon Elicitation at 24 h

At T 24 h, we observed changes to distribution of putative functions of DAPs compared to T 7 h (Figure 4). In shoots treated locally by foliar spraying, most represented functional categories for over-accumulated proteins were DNA/RNA binding and transcriptional/translational processes (6 proteins; 20%), transport and cell wall related proteins (both 3 proteins; 10%) (Figure 4A). We described under-accumulated 3 proteins (40%) linked to carbohydrates and energy metabolism, more precisely to glycolytic processes (fructose-biphosphate aldolase [A0A078HVW3]; glyceraldehyde-3-phosphate dehydrogenase [A0A078ILF4]) and to xylose degradation (xylose isomerase [A0A078GKV7]) (Appendix A). Three proteins (40%) related to DNA/RNA binding and transcriptional/translational processes were also under-accumulated.

In shoots treated with RLs at the root level, plant defense proteins were the most abundant over-accumulated proteins (25 proteins; 25%) (Figure 4A). In a similar way to our results at T 7 h, we found that the plant defense DAPs were involved in both biotic and abiotic stress responses (Appendix A). The rest of the over-accumulated proteins were distributed into diverse categories such as DNA/RNA binding and transcriptional/translational processes (11 proteins; 11%), secondary metabolism (7 proteins; 7%) and transport (6 proteins; 6%) (Figure 4A). 

The category DNA/RNA binding and transcriptional/translational processes was the most represented functional class among under-accumulated proteins (8 proteins; 18%), followed by transport (5 proteins; 11%), cell wall related proteins and carbohydrates/energy metabolism (4 proteins each; 9%). The latter category included an alpha amylase (A0A078FEH0) and glycolytic processes-related proteins (fructose-biphosphate aldolase [A0A078HVW3]; pyruvate kinase [A0A078JMR1]) (Appendix A).

Cell wall related DAPs included proteins participating in primary cell wall rearrangement such as a pectin acetylesterase (A0A078F857), a pectate lyase (A0A078GH88) and xyloglucan endotransglucosylase/hydrolases (A0A078H6P7; A0A078F5R5) (Appendix A). 

The distribution of the DAPs function in roots after RL treatment by foliar spray showed variations at T 24 h compared to T 7 h (Figure 4B). Transport was the most represented over-accumulated proteins category (4 proteins; 27%) with other categories evenly represented with 1 or 2 proteins each. Interestingly, photosynthetic processes proteins were the most abundant among the under-accumulated proteins (10 proteins; 21%), with DAPs such as photosystem I reaction center subunit proteins, photosystem II protein D1, chlorophyll a-b binding proteins and oxygen-evolving enhancer proteins (Appendix A). Four under-accumulated proteins (8%) were described for the following classes: carbohydrates and energy metabolism, cell wall metabolism, plant defense/stress. 

After local RL application on roots for 24 h, no big change in the distribution of functions of DAPs was apparent compared to results at T 7 h. Plant defense/stress and secondary metabolism were still the most represented functional category for over-accumulated proteins with 22 (17%) and 17 (13%) proteins, respectively (Figure 4B). Over-accumulated proteins linked to ROS scavenging/redox homeostasis (17 proteins; 13%) and to cell wall metabolism (11 proteins; 8%) were more abundant at T 24 h compared to T 7 h. Over-accumulated proteins involved in the response to oxidative stress were associated with the glutathione-ascorbate cycle and glutathione peroxidase pathway, such as peroxidases, glutathione transferases, catalase and thioredoxin reductase (Appendix A). 

Most represented functional classes for under-accumulated proteins in roots treated by root absorption at T 24 h were identical to the ones identified at T 7 h and were as followed: DNA/RNA binding and transcription/translation (39 proteins; 24%), transport and plant defense/stress (21 proteins each; 13%) (Figure 4B). Notably, the polyadenylate-binding protein RBP45B (A0A078HAY9) had its abundance severely depleted (FC of 0.01 at T 24 h compared to control) (Appendix A). 

## 3. Discussion

In the present work, a comparative quantitative proteomic analysis was carried out to evaluate the effects of two types of RL treatment, foliar spray or root absorption, during 2 time points (T 7 h and T 24 h) on shoot and root proteomes in rapeseed. Indeed, previous proteomic studies focusing on effects of well-described elicitors, have reported significant modifications on plant proteomes upon elicitation [34,35,36,37]. Our results show that RLs affect protein abundance in both shoots and roots and that the type of application influences protein accumulation. To be precise, we have found that RL treatment by root absorption led to more DAPs in both shoots and roots compared to foliar spray. Additionally, modifications in protein abundance were higher at T 24 h than at T 7 h. 

Furthermore, the present study allowed us to investigate the potential differences between the plant local response and systemic response upon RL elicitation. Our results show that RL application modify protein abundance levels in shoots and roots upon both types of application (foliar spray or root absorption), triggering local and systemic responses at the protein level. To our knowledge, this is the first report of RLs potentially activating systemic defense mechanisms in plants at the proteomic level. Previous studies have shown that the perception of microbial or pathogen-associated molecular patterns (MAMPs or PAMPs) by plants triggers the set-up of systemic resistance. This elicitor-triggered resistance is regulated by signaling pathways involving major phytohormones such as salicylic acid (SA), jasmonic acid (JA) and ethylene (ET) and initiates the biosynthesis of secondary metabolites and other defense molecules [4,38]. Such elicitors include the flg22 peptide composed of 22 amino acids from a conserved region of the N-terminus of the bacterial flagellin which is known to trigger strong local defense responses as well as systemic responses [39]. Other elicitors such as oligosaccharides [40] or lipopeptides [41], have been shown to activate plant defense mechanisms and systemic protection against a wide range of pathogens. Lipopeptides, which are secreted microbial biosurfactants similar to RLs, can activate the Induced Systemic Resistance (ISR), although their recognition by plant cells and the way they activate ISR remain unclear [13,42].

Previously, RLs have been shown to trigger local protection against various pathogens in grapevine, *A. thaliana*, and *B. napus* [20,21,23]. Early signaling responses triggered by RLs include accumulation of ROS [20,21,22,23], calcium influx and MAP kinase activation [20]. Other defense responses consist of callose deposition, hormone production, defense gene activation and hypersensitive reaction-like response [20,21,22,23]. Moreover, different signaling pathways are involved in local RL-mediated resistance depending on the type of pathogen [21].

In our study, we have shown how RL elicitation also affects de novo protein synthesis associated with plant defense, antioxidant systems, signaling pathways, secondary metabolism and cell wall modification, at local and systemic levels in shoots and roots of rapeseed, giving insight on RLs being potential inducers of plant systemic resistance. Nevertheless, this would remain to be established by performing phytoprotection studies.

Our results highlight a clear effect of the type of RL application on protein abundance in rapeseed. Globally, we have observed stronger local and systemic responses after RL application on roots compared to RL application on shoots at both time points (T 7 h and T 24 h). Root absorption treatment also largely impacts protein abundance in shoots, suggesting a significant systemic response induced by this type of treatment. The higher response observed in roots as compared to shoots treated locally is probably linked to the tissue nature of the different organs, but it could also be related to the natural distribution of *P. aeruginosa*, which is more present in soil than in plant tissues [43] and could explain a better root sensitivity to RLs. On the contrary, foliar spraying led to few changes in protein abundance in shoots. Interestingly, more DAPs were identified in roots compared to shoots after foliar spraying, suggesting that the place of application has a crucial role in the plant response. Additionally, clear differences in rapeseed shoot and root proteomes were visible upon RL treatment, with most DAPs being specific to one type of RL application. Previous works have described different response strategies put in place depending on the organ. Roots are a privileged site of entry for pathogens and can activate defense mechanisms in response to various elicitors. Specifically, clear differences in plant defense strategies between shoots and roots have been highlighted in recent years, with immune responses appearing to be specific and compartmentalized in roots [44]. 

To further investigate the differences between rapeseed shoot and root responses upon RL application, we have classified the DAPs into categories related to biological processes according to their putative functions. 

On one hand, we have found that functional categories associated with primary metabolism such as photosynthesis, carbohydrates, amino acid, and energy metabolism were minorly represented among the DAPs showing a low effect of RLs on these processes. Under microbial elicitation, plant primary metabolism is repressed [37,45]. Our data shows that there is overall no major under-accumulation of proteins related to primary metabolism pathways. This result suggests that RLs are elicitors with low cost on plant fitness and energy, which could constitute an advantage for their use for biocontrol purposes. On the other hand, RL application is responsible for a large over-accumulation of plant defense proteins especially in roots. Putative functions of these types of DAPs include response to both abiotic stress (HSPs and LEA proteins) and biotic stress. DAPs related to biotic stress response include pathogenesis-related proteins (PR proteins), enzymes with chitinase activity and ankyrin repeat-containing proteins while DAPs linked to abiotic stress response include proteins involved in defense against cold/heat or drought. Although the over-accumulation of abiotic defense proteins might seem surprising upon bacterial eliciting, several studies have previously shown similarity between the plant response after RL treatment and abiotic stress responses in plants. Accumulation of metabolites involved in the response to abiotic stress after synthetic RL treatment has also been described in wheat [46]. These results could be linked to the potential direct interaction between RLs and the plant plasma membrane, triggering downstream defense reactions as observed in abiotic stress-like responses. Moreover, our functional analysis highlighted a few modifications in phospholipid metabolism, which could also be related to the RL direct interaction with the plasma membrane. In our work, we have conducted an extraction of total proteins from roots and aerial parts of rapeseed for comparative proteomic analysis. Performing a plasma membrane protein extraction to study the effect of RLs on plasma membrane proteins could provide a better understanding of the perception of RLs by the plant cells, as it is currently not well described.

According to our results, activation of secondary metabolism also plays a large part in both local and systemic plant response to RL elicitation. Secondary metabolism refers to metabolic pathways leading to the production of molecules that are considered non-essential for the primary functions of plant organisms such as growth and reproduction [47]. These molecules can be classified into different molecular families such as phenolics, terpenes, alkaloids, flavonoids, and steroids [48]. Plant secondary metabolites play many roles, mainly as mediators during plant-environment interactions, although it has been shown they can also be integrated into primary metabolic networks [49]. It has been shown that elicitors such as flg22 and oligogalacturonides (OGs) positively regulate key genes of secondary metabolism involved in the biosynthesis of glucosinolates, camalexin and phenylpropanoids such as cytochrome P450 (CYP) and PAL genes [45]. Recent metabolomics analysis also showed positive impact of flg22 treatment on secondary metabolites production in tomato [50]. 

Our results have shown a large amount of over-accumulated proteins involved in secondary metabolism, especially proteins linked to flavonoids and glucosinolates metabolism. Flavonoids are ubiquitous metabolites which serve many functions in plants. They participate in plant protection against biotic and abiotic stresses such as UV radiation [51]. Flavonoids can also act as signaling molecules as well as detoxifying and antimicrobial compounds. Glucosinolates are sulfur-containing secondary metabolites produced by plants from the *Brassicaceae* family [52]. These molecules participate in plant defense against herbivores [53]. It is then plausible that the use of RLs as plant defense elicitors on rapeseed is responsible for modifications on glucosinolates metabolism.

Another functional category related to plant defense which was significantly represented among over-accumulated proteins is ROS scavenging enzymes/redox homeostasis. ROS are key molecules in signaling and for induction of plant defenses [54]. Their excessive production caused by biotic and abiotic stress can lead to oxidative damage. Other elicitors such as oligosaccharides have been shown to play a role in stimulation of antioxidant proteins synthesis [40,55]. Here, we have shown that proteins involved in the glutathione-ascorbate cycle and glutathione peroxidase pathway such as peroxidases, glutathione transferases and catalase were over-accumulated upon RL treatment, in roots particularly. Monnier et al. (2018) previously showed that RLs also activate early responses by triggering ROS production in foliar disks of *B. napus*. Our work also suggests that RLs stimulate late responses to counteract cellular oxidative stress caused by ROS, such as de novo synthesis of antioxidant proteins. 

A crucial component of the set-up of the systemic plant defense response is phytohormones and related signaling pathways. In our study, we have identified differentially accumulated signaling and phytohormones-related proteins depending on the organ and type of RL application. In roots especially, over-accumulated proteins such as 12-oxophytodienoate reductases (OPRs) played a role in the JA pathway [34,35,45] whereas prominently under-accumulated proteins such as aminocyclopropane-1-carboxylate oxidase (ACC oxidase) were linked to ET biosynthesis [37]. 

RLs have been shown to trigger defense gene activation and signaling molecules accumulation depending on the type of pathogen. ET is involved in RL-induced resistance to biotrophic and hemibiotrophic pathogens, while JA is essential for resistance against the necrotrophic fungus *B. cinerea* [21]. Although SA is described to play an essential role for RL-mediated resistance to all types of pathogens [21] and RLs have been described to trigger the *BnPR1*, gene [23], it is noticeable that we did not report any change in abundance of proteins related to the SA pathway in our proteomics study. Indeed *PR1* gene is a well-known SA pathway defense gene marker [56,57]. This could be due to the different treatment conditions used in the different studies but also show the importance of performing protein analyses to study the elicitor effects on plants.

The modifications in protein accumulation associated with hormonal signaling pathways that we have described in our study suggest that RLs could stimulate defense responses in distant tissues by triggering hormonal signal transduction. 

Altogether, our results suggest that there is a different effect of the type of RL application (foliar spraying or root absorption) on shoots and roots of *B. napus*. Using the TMTs labeling approach, we identified proteins that are synthesized upon different RL treatments, thus allowing a better understanding of molecular responses of *B. napus* depending on the type of elicitor application. As the level of protection provided by elicitors is pathogen-dependent, this work could help to design further studies focusing on the use of RLs as elicitors against specific rapeseed pathogens, thus contributing to the development of biocontrol strategies and a sustainable agriculture. 

## 4. Materials and Methods

### 4.1. Biological Materials and Culture Conditions

Seeds of *Brassica napus* cultivar Darmor-*bzh* were harvested in July 2019 and kept in the dark at 4 °C. Seeds were sterilized according to [22]. After 7 days, the seeds were placed on plant growth medium MS [Murashige and Skoog basal medium (Sigma–Aldrich, St. Louis, MO, USA), 0.5 g L^−1^ MES (2-(*N*-morpholino)ethanesulfonic acid), and 5 g L^−1^ sucrose] solidified with agar (7 g L^−1^) and equilibrated to pH 5.7. The seeds were placed for 10 days in a climatic chamber at 21 °C, 60% relative humidity, 150 μmol m^−2^ s^−1^ light intensity with a 16 h photoperiod. Ten-day-old seedlings were transferred to 4.3 g L^−1^ MS liquid medium with 0.5 g ^L−1^ MES, pH 5.7. The tubes containing the seedlings were placed for 24 h under see-through plastic in the climatic chamber at 22 °C the day and 18 °C the night, 45% relative humidity with a 16 h photoperiod.

### 4.2. Preparation of RL Solutions and Applications

The RL solutions were prepared at 0.1 g L^−1^ from a 90% RLs mix (reference R90-50G, AGAE Technologies, Corvalis, USA). The mix was previously analyzed and composed of 66% mono-RLs and 34% di-RLs. The two major compounds were mono- and di-RLs with two saturated C10 fatty acid chains [23]. 

Before treatment, plantlets were placed in MS liquid medium during 24 h for acclimatation. For RL treatment by foliar spraying to run-off, RLs were dissolved in sterile water and sprayed on cotyledons and leaves at the 2-leaf stage. For RL treatment by root absorption, RLs were first dissolved in water and added in the MS liquid medium in which the plantlets were transferred with roots totally immersed in the medium. Thirty plantlets were collected per treatment condition (without treatment, RL treatment by foliar spraying and RL treatment by root absorption). The shoots, corresponding to all the above ground mass, and the roots were harvested separately after 7 h or 24 h of treatment. Samples were immediately frozen in liquid nitrogen and stored at −80 °C. Samples were then ground to fine powder in a ball mill and stored at −80 °C again until proteomic analysis.

### 4.3. Proteomic Analysis

Proteins were extracted from the shoots and roots using iST sample preparation kit (Preomics, Planegg (Martinsried), Germany) using manufacturer’s recommendations. Proteins were identified and quantified using a Tandem Mass Tags (TMTs)-based comparative proteomics analysis method [58]. These experiments were performed on three biological replicates. Data were processed using Proteome Discoverer 2.4 (ThermoFisher Scientific, Bremen, Germany) before being run against *Brassica napus* TrEmbl database (release 2022_04). The MS/MS data (raw data, identification and quantification results) are available to ProteomeXchange Consortium (http://www.proteomexchange.org (accessed on 4 December 2022) *via* the PRIDE partner repository with dataset identifier PXD038531 an [59].

To identify DAPs, the results were classified according to the ratio of proteins amounts in shoots treated with RLs vs. control and roots treated with RLs vs. control at T7 (7 h of treatment), T24 (24 h of treatment) according to foliar spraying and root application. DAPs were defined as proteins with a fold change >1.7 or <0.6 at a *p*-value <0.05 between two comparison groups (shoot-RLs-T7-foliar spraying vs. shoot without RLs-T7, root-RLs-T7-foliar spraying vs. root without RLs-T7, shoot-RLs-T7-root application vs. shoot without RLs-T7, root-RLs-T7-root application vs. root without RLs-T7). The same comparison groups were analyzed for the 24 h treatments.

### 4.4. Functional Classification

Functional annotation of DAPs was performed using Blast2GO (https://www.blast2go.com (accessed on 13 October 2022) and Uniprot database (https://www.uniprot.org (accessed on 12 October 2022). The peptides sequences of all DAPs were extracted and submitted to NCBI for BLAST search (https://blast.ncbi.nih.gov/Blast.cgi (accessed on 14 October 2022) using default parameters, with *Brassica* as the organism filter. The protein families were also reclassified according to literature information.

## Figures and Tables

**Figure 1 ijms-24-02390-f001:**
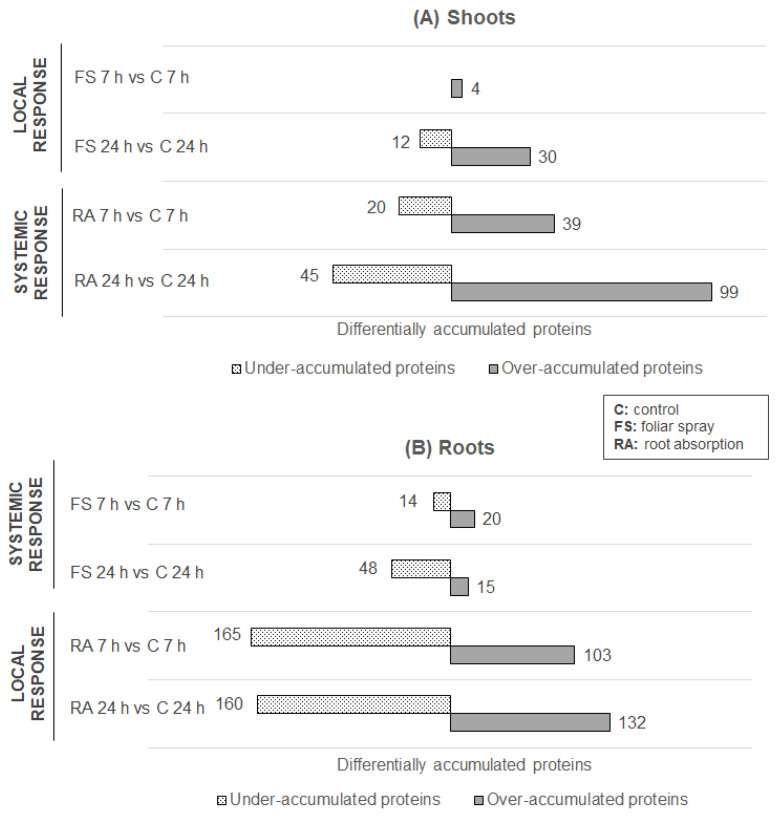
Differentially accumulated proteins (DAPs) in shoots (**A**) and roots (**B**) of *B. napus* var. Darmor-*bzh* treated with RLs versus control. Four RL treatment conditions were tested: foliar spray treatment (FS) and root absorption treatment (RA), during 7 h or 24 h. The DAPs are identified according to Blast2GO (https://www.blast2go.com (accessed on 13 October 2022)) and Uniprot database (https://www.uniprot.org (accessed on 12 October 2022). These experiments were performed on three biological replicates.

**Figure 2 ijms-24-02390-f002:**
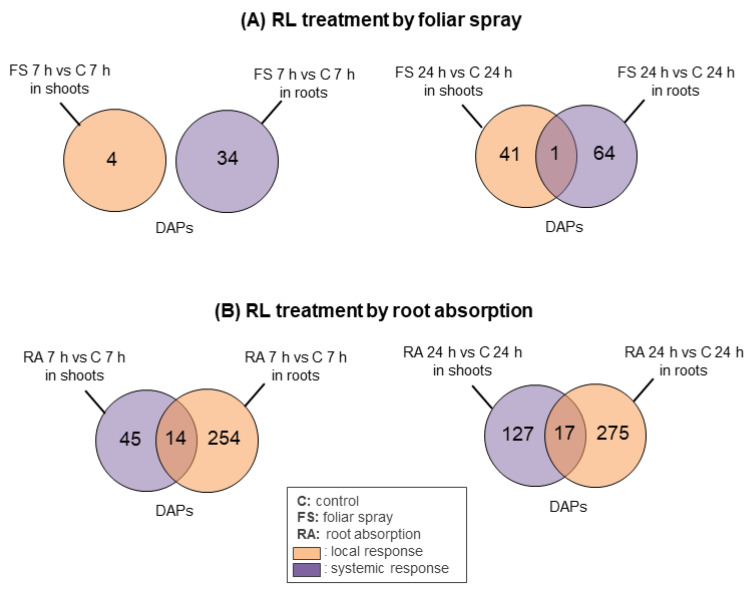
Venn diagrams of differentially accumulated proteins (DAPs) in shoots and roots of *B. napus* var. Darmor-*bzh* treated with RLs versus control. Four RL treatment conditions were tested: foliar spray treatment (**A**) and root absorption treatment (**B**), during 7 h or 24 h. These experiments were performed on three biological replicates.

**Figure 3 ijms-24-02390-f003:**
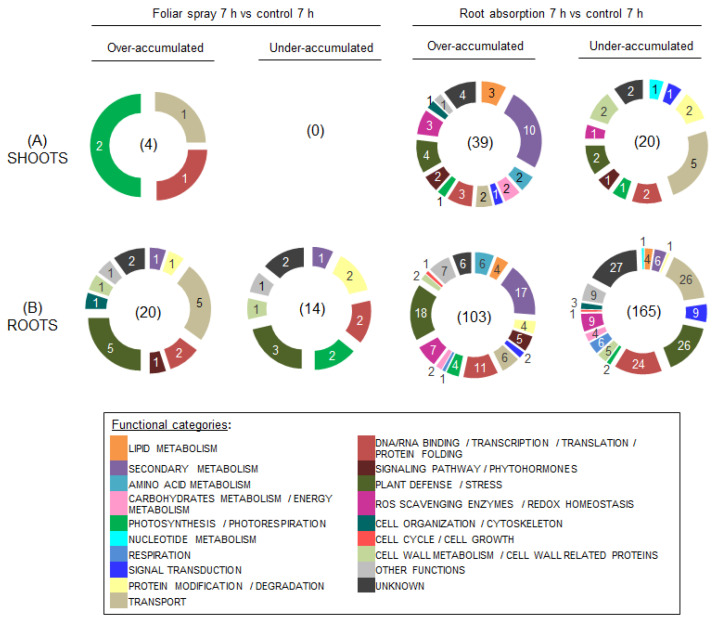
Functional classification of differentially accumulated proteins (DAPs) in shoots (**A**) and roots (**B**) of *B. napus* var. Darmor-*bzh* treated with RLs by foliar spray or root absorption versus control during 7 h. The protein families are assigned based on the information available in the Uniprot and Gene Ontology databases and reclassified according to the literature information. These experiments were performed on three biological replicates.

**Figure 4 ijms-24-02390-f004:**
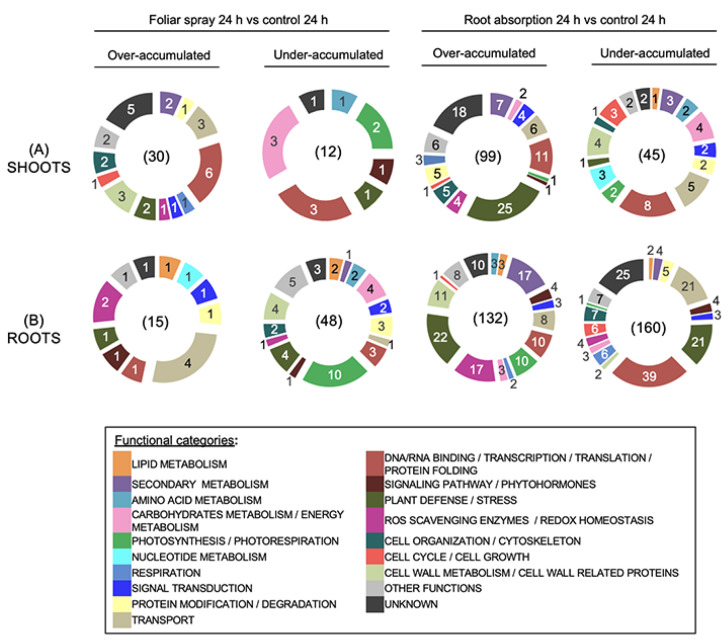
Functional classification of differentially accumulated proteins (DAPs) in shoots (**A**) and roots (**B**) of *B. napus* var. Darmor-*bzh* treated with RLs by foliar spray or root absorption versus control during 24 h. The protein families are assigned based on the information available in the Uniprot and Gene Ontology databases and reclassified according to the literature information. These experiments were performed on three biological replicates.

## Data Availability

Our proteomic dataset is submitted to ProteomeXchange via the PRIDE database (see Material and Methods). The project name is “Impact of rhamnolipids (RLs), natural defense elicitors, on shoot and root proteomes of Brassica napus by a Tandem Mass Tags (TMTs) labeling approach”. The project accession is PXD038531 and the project DOI is 10.6019/PXD038531.

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
