# Peer review of "Impact of Rhamnolipids (RLs), Natural Defense Elicitors, on Shoot and Root Proteomes of Brassica napus by a Tandem Mass Tags (TMTs) Labeling Approach"

_ijms, 2023, doi:10.3390/ijms24032390_

Round 1
Reviewer 1 Report
Similar to so many reported proteomics studies, this study also produced extensive data and provided some new and interesting information about the response of plants to the application of RLs. However, there are still some deficiencies in the experimental design, I think. For example, if you want to know which application way for RLs is better, you should treat the infected plants, instead of the healthy plants, and compare the effect of the RLs treatment that applicated in a different way.
Author Response
We thank both reviewers for their careful reading of our manuscript. In the following, we address the points raised that helped to improve the manuscript. Corrections and suggestions corresponding to Reviewer 1 comments are in blue in the main document, corrections and suggestions corresponding to Reviewer 2 are in red.
Reviewer 1
Similar to so many reported proteomics studies, this study also produced extensive data and provided some new and interesting information about the response of plants to the application of RLs. However, there are still some deficiencies in the experimental design, I think. For example, if you want to know which application way for RLs is better, you should treat the infected plants, instead of the healthy plants, and compare the effect of the RLs treatment that applicated in a different way.
We agree with the Reviewer 1 comment. Nevertheless, the main objective of the article was to characterize the eliciting effects of RLs at the proteomic level considering the mode of application and the answers locally or distally. It was not to define the best way of application. As suggested, to better define the strategy of application of the RLs to protect rapeseed from a specific pathogen, assays with the pathogen would be needed.
To clarify it and avoid any confusion for the reader we have made some changes in the text.
-line 89: we have removed the mention to aerial or root pathogens
-line 344: After “RLs being potential inducers of plant systemic resistance”, we have added the sentence “Nevertheless, this would remain to be established by performing phytoprotection studies.”
-lines 455 to 458: We have replaced “These results could help to define the best way of RL applying depending on the targeted pathogen to contribute to the development of biocontrol strategies and a sustainable agriculture.” By “As the level of protection provided by elicitors is pathogen-dependent, this work could help to design further studies focusing on the use of RLs as elicitors against specific rapeseed pathogens, thus contributing to the development of biocontrol strategies and a sustainable agriculture.”
Reviewer 2 Report
Please see the attached file

Author Response
We thank both reviewers for their careful reading of our manuscript. In the following, we address the points raised that helped to improve the manuscript. Corrections and suggestions corresponding to Reviewer 1 comments are in blue in the main document, corrections and suggestions corresponding to Reviewer 2 are in red.
Reviewer 2
The manuscript titled “Impact of rhamnolipids (RLs), natural defense elicitors, on shoot and root proteomes of Brassica napus by a Tandem Mass Tags (TMTs) labeling approach”, by Pierre et al., is a very promising and novel study that looked at the proteomic coverage of B. napus plants in response to RL application. The study extensively investigated the proteins that were potentially influenced by the treatment. The manuscript is detailed and very well written. Great work!
Couple of questions
- A bit confused about the number of plantlets used in the study. The manuscript mentions 30 plantlets were collected. Is it 30 each of control, and treated or 30 plantlets in total (15 control/15 treated)? If it is 15 each, then the data present here is for 7 shoots and 7 roots (n=7)? The section about the preparation and application needs to be explained in detail, either in the main text or in the Supporting information.
We agree that it was confusing. Thirty plantlets per treatment condition were collected. The data presented here are for 30 shoots and 30 roots for each treatment.
The clarification has been made in the text: line 488 to 490.
- How was the RL solution prepared? What was the composition of the RL mix. The reference R90 from AGAE Tech cited in the paper cannot be found on the website. The authors referenced a paper by Monnier et al. which defined the RL mix types, but sourced from a different vendor/Lab. It will be good to list the RL mix composition in the manuscript
The precisions have been added in the text: line 477 to line 480. The RL reference was R90-50G, from AGAE Technologies (Corvalis, USA). It seems that the provider has recently changed the references in its catalog. As we explain, we have previously analyzed the mix. We have clarified it in the paper and added that it is composed of 66% mono-RLs and 34% di-RLs and that the two major compounds were mono- and di-RLs with two saturated C10 fatty acid chains. There was a mistake in the reference of the article and we have corrected it.
- How was the RL solution prepared? Was it diluted in water? or a different solvent, as it is a lipid? If a different solvent, was the control plants treated with the identical solvent, to account for delivery control?
For foliar treatment, RLs were dissolved in sterile water. Indeed, RLs are amphiphilic molecules soluble in water. For root treatment, RLs were first dissolved in water and added in MS liquid medium.
The changes have been made in the text: line 484 to line 486.
- For the foliar application of the RL, how was the treatment applied? Was the RL mix sprayed on the leaves? During the application stage what was the foliar coverage or how many leaves were there to ensure uniform coverage of the treatment application?
For foliar spraying, RLs were sprayed on cotyledons and leaves at the 2-leaf stage to run-off.
The changes have been made in the text: line 483 and line 484.
- When the authors mention shoot were harvested and used for analysis, does it include all the above ground mass (leaves, meristem, shoot/stem)
Shoots included all the above ground mass.
We specified it: line 491 to line 492.
- Although not necessary for this manuscript, out of curiosity, did the authors measure the RL that was left in the solution? Which component of the RL mix could be influential in eliciting the observed responses?
We didn't measure the RLs that were left in the solution. We do not know exactly the specific eliciting influence of each RLs in the mix. A study, on synthetic Mono-Rhamnolipids, has shown that the C10 (ten carbons in the fatty acid chains) or C12 forms are stronger to trigger ROS production (https://www.mdpi.com/1420-3049/25/14/3108). Another study has shown that mono- and di-rhamnolipids from P. aeruginosa used separately are active on grapevine cells (https://onlinelibrary.wiley.com/doi/10.1111/j.1365-3040.2008.01911.x). To decipher specifically the part played by each compound of the mix in the eliciting activity would be another interesting study. But, our objective here, was to investigate the global effects of the RLs from the secretome of the bacteria Pseudomonas aeruginosa, being the main producer of RLs.
Minor comment
Ln 463: Correct the words “manufactor’s recommandations” to manufacturer’s recommendations
Done in the manuscript: line 498.